# Therapeutic Compliance of Patients with Arterial Hypertension in Primary Care

**DOI:** 10.3390/medicina56110631

**Published:** 2020-11-22

**Authors:** Mihaela Adela Iancu, Irina-Ioana Mateiciuc, Ana-Maria Alexandra Stanescu, Dumitru Matei, Camelia Cristina Diaconu

**Affiliations:** 1Department 5, “Carol Davila” University of Medicine and Pharmacy, 050474 Bucharest, Romania; irinamateiciuc@yahoo.com (I.-I.M.); alexandrazotta@yahoo.com (A.-M.A.S.); drmateidumitru@yahoo.com (D.M.); drcameliadiaconu@gmail.com (C.C.D.); 2Primary Care Office, 011165 Bucharest, Romania; 3Alessandrescu-Rusescu National Institute for Mother and Child Health, 020395 Bucharest, Romania; 4Department of Internal Medicine, Clinical Emergency Hospital of Bucharest, 105402 Bucharest, Romania

**Keywords:** compliance to treatment, arterial hypertension, comorbidities, general practitioner

## Abstract

*Background and Objectives:* Arterial hypertension remains an important cause of cardiovascular morbidity and mortality, despite all the progress made in the methods of diagnosis, monitoring of target organs’ damage and treatment. The main cause of the increased prevalence of uncontrolled blood pressure values is the low compliance to antihypertensive treatment. The objective of our study was to assess the compliance to the treatment of patients diagnosed with arterial hypertension and monitored in a primary care office. *Materials and Methods:* The cross-sectional, retrospective study included 129 patients, 65.89% (85) women, previously diagnosed with arterial hypertension. Data from the medical files were analyzed, as well as the patients’ answers to a survey of 18 questions regarding arterial hypertension, comorbidities, complications, treatment and awareness of the condition. *Results:* The study included 129 patients, with a mean age of 66 ± 8 years. The majority of patients were overweight, 55.81% (72 patients), and 10.85% (14 patients) had grade I obesity. Most of the patients, 55.81% (72 patients) were diagnosed with grade III hypertension, while 37.98% (49 patients) were diagnosed with grade II hypertension and 6.2% (8 patients) with grade I hypertension. One third of the surveyed patients answered that they follow the recommendations of a low-sodium diet, 21.7% are adherent to treatment, but 56% think that the total cost of the medication is an impediment for their compliance to treatment. The majority, 82.17% (106 patients), of respondents had an affirmative answer to the questions: ‘Do you think it would be easier to take one pill instead of 2, 3 or 4 pills?’ *Conclusion:* The increased compliance to the antihypertensive treatment and control of blood pressure values are associated with the degree of awareness of arterial hypertension and the consequences if left untreated, emphasizing the role of the general practitioner in counseling for secondary prevention.

## 1. Introduction

Arterial hypertension represents the most widely spread cardiovascular disease (CVD) in the world (1.13 billion people). As life expectancy rises, there are increasingly more individuals with sedentary lifestyle. As a consequence, the prevalence of hypertension worldwide will continue to rise [1]. It is estimated that the number of people suffering from arterial hypertension will increase by 15–20% by 2025, reaching almost 1.5 billion [2]. In Europe, arterial hypertension has a higher prevalence (60%) compared to the USA (46%) or Canada (32–46%) [3]. It is recognized that arterial hypertension is responsible for approximately 25% of myocardial infarction cases and it is the cause of 42% of the annual deaths in Europe [4].

In Romania, the prevalence of arterial hypertension in the adult population in 2018 was estimated at 44.1%, meaning approximately 7.5 million Romanians were suffering from arterial hypertension in the year 2016. Only 81% of them knew about their condition, as revealed in the Study of the Prevalence of High Blood Pressure and Cardiovascular risk assessment in Romania (SEPHAR III study), which places Romania on the top position on Europe’s risk map [5].

SEPHAR III, conducted in 2016, proposed an improvement of the previous study (SEPHAR II) by introducing paraclinical investigations that allowed a complete evaluation of the cardiovascular risk, and by annually monitoring the patients included in the study, throughout follow-up check-ups (‘SEPHAR follow-up’) [5].

According to SEPHAR III, approximately 1 in 5 Romanians is not aware of his diagnosis of arterial hypertension [5,6]. One of the aspects that stands out in the SEPHAR III study is the increase of the awareness of high blood pressure, from 44% in 2005 to 81% in 2016. Also, the compliance to antihypertensive treatment has increased from 39% in 2005 to 75% in 2016, with the estimation of an increase to 91% by 2020 [5]. According to the same study, the rate of therapeutic control has seen an increase, from 20% in 2005 to approximately 31% in 2016, with the estimation that by 2020, 36% of the patients suffering from arterial hypertension will have their blood pressure under control [5].

Despite the recommendations of the current guidelines and the efforts made to develop an algorithm for an optimal approach to arterial hypertension, of the progress made in monitoring the patients and pharmacological treatments, only 40% of hypertensive patients have controlled blood pressure values [7]. The lack of arterial hypertension control in two thirds of the patients has raised the issue of the necessity for a better therapeutic strategy. In the attempt of finding an explanation for the low overall rate of arterial hypertension control, the question of the effectiveness of treatment was not raised, but rather the lifestyle and the low compliance to treatment of the patients diagnosed with arterial hypertension [8]. Recent studies have proven that the greater the increase in the number of drugs prescribed, the lower the compliance rate of the patients [9]. Measuring patients’ compliance to treatment based on the drug level in blood and urine and the frequency of the prescription’s renewal led to the conclusion that the complexity of the prescribed treatment has a negative influence on the long-term compliance rate [10].

Compliance was defined as the patient’s behavior following the recommendations regarding medication, diet, lifestyle changes and future medical appointments. Adherence is the extent to which a patient participates in a treatment regimen after agreeing to given recommendations [11]. Taking into account the recommendation of the current guidelines to achieve stricter therapeutic targets, the strategy considered optimal in addressing arterial hypertension encourages the use of fixed combinations or a fixed dosage combination [12,13]. Other factors related to the patient that can determine a lower compliance to the arterial hypertension treatment are unwillingness to accept the diagnosis, lack of awareness of potentially fatal consequences of not treating arterial hypertension (especially in the context of its asymptomatic phase), fear of side effects or preference for alternative therapy [14]. Regarding lifestyle, and especially compliance with the low-sodium diet, an underestimation of the amount of salt contained by some products (which generates unintentional non-compliance), especially for processed products, which contain 80% of the salt consumed, was frequently noticed [1,15].

The fact that high blood pressure is a condition which is mainly addressed by the primary medical assistance emphasizes the general practitioner’s role in obtaining a better compliance to treatment; thus, a higher compliance to treatment was highlighted once the number of healthcare providers increased, with the increase attributed to easier access to healthcare but also to a more rigorous follow-up of hypertensive patients [16,17].

Beyond simply prescribing the treatment, the physician’s ability in promoting a healthy lifestyle, involving the patients in the therapeutic decisions and helping them overcome their unjustified fears, can improve compliance to treatment [14,18]. Thus, effective communication and a doctor–patient relationship based on trust have a positive influence on the compliance to treatment, while the burnout phenomenon to which those working in healthcare are exposed can have a negative impact on compliance.

The objective of the study was to assess the compliance to treatment of patients diagnosed with arterial hypertension, in the records of a general practitioner. We aimed to assess the compliance to non-pharmacological and pharmacological treatment of patients diagnosed with arterial hypertension, to identify the control rate of blood pressure values and to evaluate the degree of disease awareness.

## 2. Materials and Methods

The cross-sectional, retrospective study was conducted between September 2019 and October 2019 in a primary care office from Bucharest, Romania. It included 129 patients, aged between 43 and 81 years old, previously diagnosed with high blood pressure, undergoing treatment and being in the records of the office. The patients were enrolled in the study in the order in which they visited the physician’s office for monitoring and treatment.

The data were collected from medical files and based on the answers the patients gave to a survey of 18 questions, referring to arterial hypertension, comorbidities, complications, treatment and monitoring of the condition (Appendix A). All the patients signed an informed consent to participate to the study.

In order to analyze the data, Microsoft Office Excel 2003 and IBM SPSS Statistics 20 were used. A *p*-value less than 0.05 (≤0.05) is statistically significant.

## 3. Results

The study included 129 patients, aged between 43 and 81 years old, with the mean age 66 ± 8 years. Among them, 34.10% (44) were men and 65.89% (85) women (Table 1). The analysis of their living environment showed a predominance of the urban area, 80.62% (104), compared to the rural area, 19.37% (25). The distribution by age reveals the predominance of those aged between 60 and 69 years old, followed by those aged between 70 and 79 years old, supporting the increase in the prevalence of arterial hypertension along with aging (Figure 1).

Among the risk factors involved in the onset and progression of arterial hypertension, the following were analyzed: familial history, overweightness, diabetes, dyslipidemia and smoking. The percentage of those with cardiovascular family history (stroke, acute myocardial infarction at ages less than 55 years old) and especially with a history of premature death (under 55 years old) because of cardiovascular causes was relatively low—13.95% (18 patients), the majority being represented by those without any family history.

Regarding excess weight, assessed by body mass index (BMI), most of the patients in the study group were overweight—55.81% (72 patients), 10.85% (14 patients) suffered from grade I obesity and 33.33% (43 patients) had normal weight, with the mean BMI being 28 ± 2.4 Kg/m^2^.

Regarding diabetes, only 26.35% (34 patients) of the patients included in the study had type II diabetes. Most of the patients in the study group had dyslipidemia, another major cardiovascular risk factor, present in 85.27% (110) of patients. The majority of patients included in the study suffered from mixed dyslipidemia (77.52%, 100 patients), 15.50% (20 patients) had hypercholesterolemia and 6.97% (9 patients) has hypertriglyceridemia. Another high-risk factor analyzed was smoking: 62.79% (81 patients) were non-smokers, 23.25% (30) were smokers and 13.95% (18) were ex-smokers.

The classification of patients according to blood pressure values highlighted an alarming percentage of 55.81% (72 patients) of those suffering from grade III arterial hypertension, while 37.98% (49 patients) were suffering from grade II arterial hypertension and 6.2% (8 patients) from grade I arterial hypertension (Figure 2). Regarding the onset of arterial hypertension, most of the patients, 43.41% (56), were diagnosed more than 10 years prior. 26.35% (34 patients) had a history of arterial hypertension of 6–10 years, a percentage of 26.35% (34 patients) had been suffering from high blood pressure for 1–5 years and the onset of 3.87% (8) of patients occurred less than a year before.

Within the study group, cardiovascular complications were diagnosed in a small number of patients—only 16 patients (12.4%) (*p* = 0.001). Among the cardiovascular complications, the most common was the chronic coronary syndrome—75% (12 patients) of the total complications recorded, followed by peripheral artery disease of the lower limbs—in a proportion of 18.75% (3 patients) and stroke—6.25% (1 patient).

Comorbidities have been recorded in 53 out of 129 patients (41.08%). Among those who suffered from associated conditions, most of them had digestive diseases, while the others had neurological, ophthalmologic, urologic or oncologic diseases. Out of the total number of patients with comorbidities, 38 patients (29.4% of the group) were undergoing pharmacological treatment for their comorbidities.

The assessment of hypertension-mediated organ damage (HMOD) is recommended, especially in patients with multiple cardiovascular risk factors, in order to have a better therapeutic approach and an early diagnosis of the complications. The usual investigations in assessing HMOD are blood and urine tests, electrocardiogram (ECG), cardiac ultrasound, cervical and cerebral Doppler ultrasound, Holter monitoring, fundus examination and renal ultrasound. Among patients included in the present study, 72.09% (93 patients) were investigated by blood and urine tests and ECG, 55.81% (72 patients) through cardiac ultrasound, 24.03% (31 patients) through fundus exam and 12.40% (16 patients) through Doppler ultrasound.

In the group of study, the hypertensive patients under treatment had a mean value of SBP (systolic blood pressure) of 141 ± 10 mmHg and diastolic blood pressure of 77 ± 76 mmHg.

Regarding the low-sodium diet, almost a third (29.45%, 38 patients) among those surveyed responded that they always follow it, 31.78% (41 patients) responded that they mostly follow it, 35.65% (46 patients) responded that they sometimes follow it and 3.1% (4 patients) responded that they never follow it (*p* = 0.001) (Figure 3).

The analysis of the pharmacological treatment showed that the most commonly used drugs are angiotensin-converting enzyme inhibitors (79.84%, 103 patients) and diuretics (63.56%, 82 patients), followed by beta-blockers (48.06%, 62 patients) and calcium channel blockers (44.96%, 58 patients). Within the study group, only 48 patients (37.20%) have fixed dosage combinations in their treatment regimens, while the other 81 patients (62.79%) receive separate drug classes (*p* = 0.001).

Out of the total patients surveyed, most of them (82.17%, 106 patients) answered affirmatively to the question: ‘Do you think it would be easier to take one pill instead of 2, 3 or 4?’, confirming the data of the studies which associated the use of fixed combinations with an increased compliance to treatment. To the same question, 14.72% (19 patients) answered negatively and 3.1% (4 patients) could not decide (Figure 4).

Regarding the compliance to pharmacological treatment, only 21.7% (28 patients) answered that they strictly follow treatment, without skipping any dose. 43.4% (56 patients) answered that they forget to take their treatment between 1 and 3 times a month, 15.5% (20 patients) answered that they forget to take their treatment between 4 and 7 times a month, while 19.3% (25 patients) admitted that they forget to take their treatment more than 7 times a month (*p* = 0.01) (Figure 5).

Another aspect taken into consideration by our study was the economic one—the cost of the pharmacological antihypertensive treatment. The analysis of this question’s answer showed that 56.58% (73) of the patients included in the study thought that the cost of the medication represents an obstacle in strictly following the treatment (with a statistical significance of *p* = 0.001).

The majority of the patients were aware that the treatment for arterial hypertension must be administered throughout their lifetime. When asked about the comprehension of the duration of treatment, 111 patients (86.04%) responded ‘for lifetime’, while the rest of the patients, 13.95% (18), believed that the treatment was necessary only for a limited period of time (*p* = 0.001).

A higher compliance to antihypertensive treatment and implicitly the control of blood pressure values are associated with the degree of awareness of arterial hypertension and its consequences if left untreated. When asked if they are familiar with the consequences of untreated arterial hypertension, 90.69% (117 patients) of patients answered that they are able to give at least one example, while 9.3% (12 patients) answered they are not familiar with these consequences.

According to the answers provided by the patients, 58.91% (76 patients) have an appointment with the cardiologist once a year, 12.4% (16 patients) twice a year, 1.55% (2 patients) more than twice a year, while 27.13% (35 patients) never had a cardiologist appointment.

The statistical processing revealed, based on correlation indexes, that the frequency of cardiologist appointments is influenced by the amount of time the patient has been suffering from the condition, so the patients who have an appointment more often are those who have been diagnosed with arterial hypertension for over 5 years compared to those who have been recently diagnosed. Also, the patients who have appointments more often are those who suffer from diabetes, dyslipidemia or extra-cardiovascular comorbidities (Table 2).

## 4. Discussion

Numerous studies have indicated that the compliance to antihypertensive treatment is negatively influenced by the complexity of the prescribed treatment [19], an aspect that was also confirmed by our study, in which 82.17% of patients responded that it would be easier to follow their treatment by taking just one pill instead of 2 or more.

An analysis conducted in Russia, that comprised 31 studies on the compliance to treatment, concluded that there is no association between the individual’s income and the compliance to treatment [20]. The present study indicated that for 56.58% of those surveyed (with a statistical significance of *p* = 0.001), the total cost of their prescribed medication represents an obstacle in following the treatment, not being in accordance with the Russian study. Referring to the individual’s socio-economic status, the National Health and Nutrition Examination Survey (NHANES) indicated a better compliance to treatment and a higher control rate in those with a high socio-economic status [21].

The data of the study conducted in Russia also indicates a higher compliance in the case of patients with comorbidities [20], data also supported by our study, which proved the existence of some correlations with statistical significance between diabetes, dyslipidemia, the presence of extra-cardiovascular comorbidities and the frequency of appointments to the cardiologist.

An encouraging aspect of our study is represented by the increased degree of awareness of the condition, with all 129 patients included in the study responding that they are aware of their diagnosis of arterial hypertension established by the physician. These data are in accordance with the results of the SEPHAR III study, conducted in Romania, that has determined that 555 out of 798 (approximately 70%) of those enrolled in the study were aware of their diagnosis [22]. Also, the SEPHAR III study indicated that out of the total number of patients who took part in the study, only 25% had their blood pressure under control (values under 140/90 mm Hg). Within the study group, patients showed a low rate of blood pressure values’ control, with only 34 patients (26.35%) out of 129 being under therapeutic control.

Self-monitoring of blood pressure values is considered an important factor for improving the therapeutic compliance [14]. Only 4 patients out of the total number of patients included in the study admitted that they do not monitor their blood pressure values at home.

Next to self-monitoring of the blood pressure values, the awareness of the long-term complications of arterial hypertension and the fact that the treatment must not be interrupted without the physician’s consent can influence compliance to treatment [14,18,23]. Many studies show that patients’ beliefs about the causes, and motivation to follow the therapy, were strongly related to their compliance to the treatment [24]. In our study, 90.69% of patients were able to name at least one complication of untreated arterial hypertension.

Lifestyle changes, low-sodium diet, giving up smoking, losing weight and following the prescribed treatment are factors that are also associated with a better compliance and a better arterial hypertension control rate [19,25,26]. It is possible that the low number of those who have lowered their sodium intake is not due to a low compliance, but rather to the fact that a daily intake of sodium is hard to quantify [1,14]. A good doctor–patient relationship improves compliance to the treatment [27]. In our study, the increased compliance to treatment is recorded among those who regularly have a cardiologist appointment, and the frequency of the appointments is described as a component of a compliant behavior in other studies, as well [18,28].

The current guidelines proposed, as a measure to increase the compliance to treatment and patient’s degree of satisfaction, to prescribe fixed dosage combinations [29]. This aspect was also recorded in this study. Those who have a fixed combination in their treatment regimen have a higher compliance to it.

## 5. Study Limitations

Our study presents limitations, leaving room for further in-depth analysis. The main limitations are represented by the fact that the number of patients included in the study was relatively low, as was the number of those who benefited from comorbidities’ treatment. This study does not include data about the follow-up, being unable to establish any kind of correlation between changing the treatment regimen, arterial hypertension control and compliance to treatment. Also, the fact that the data referring to the administration of the treatment were obtained through self-evaluation may represent a subjective answer, which interferes with the assessment of compliance.

## 6. Conclusions

Arterial hypertension represents a major public health issue, with an increasing prevalence. The study showed the increase of arterial hypertension prevalence among overweight patients, which is correlated with the presence of comorbidities.

The study proved that an increased awareness of the presence and consequences of hypertension revealed the existence of some correlations with statistical significance between the presence of type II diabetes, dyslipidemia, extra-cardiovascular comorbidities and the frequency of cardiologist appointments. Most of the patients are aware that the duration of their treatment will be continuous throughout their lifetime, being sensitized to the fact that even though hypertension can be asymptomatic, the treatment will still last for a lifetime. Romanian patients who regularly visit their general practitioner’s office and have a relationship based on trust with their physician have a slightly increased degree of compliance to treatment. In order to improve the compliance to treatment, the general practitioner may recommend a personalized treatment to each patient, both depending on the presence of cardiovascular risk factors and comorbidities, as well as on the individual, family, educational and socioeconomic factors. Through counseling and active monitoring, the general practitioner may contribute to a better awareness of the condition and a higher compliance to antihypertensive treatment. We suggest general practitioners to intervene early through counseling on the individualized consequences of non-compliance to treatment, as well as by adapting the treatment regimen to individual factors.

## Figures and Tables

**Figure 1 medicina-56-00631-f001:**
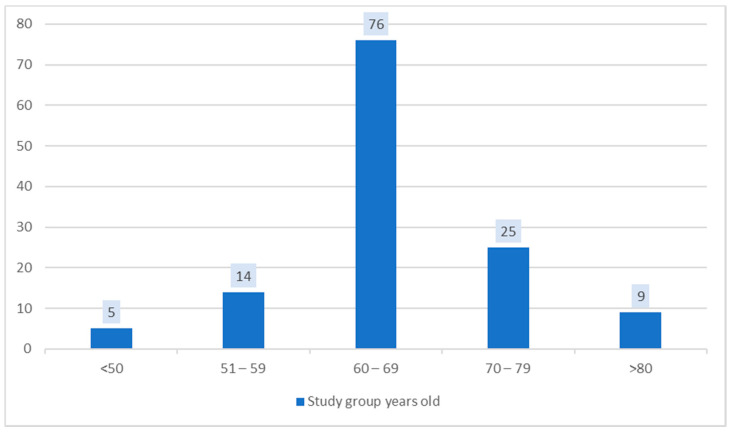
Distribution by age of the study group.

**Figure 2 medicina-56-00631-f002:**
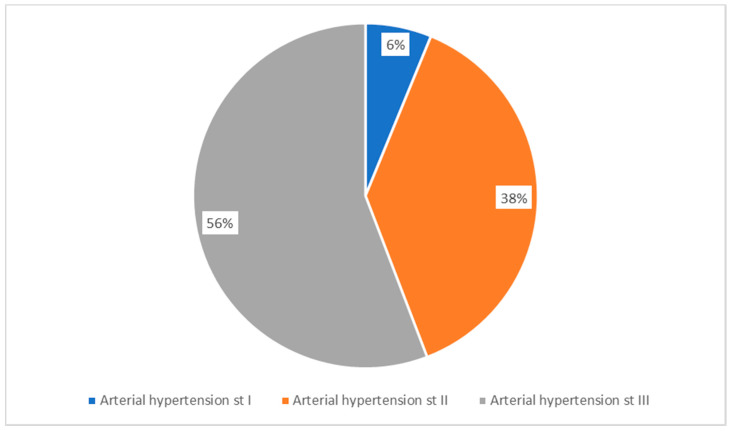
Patients’ distribution according to arterial hypertension stage.

**Figure 3 medicina-56-00631-f003:**
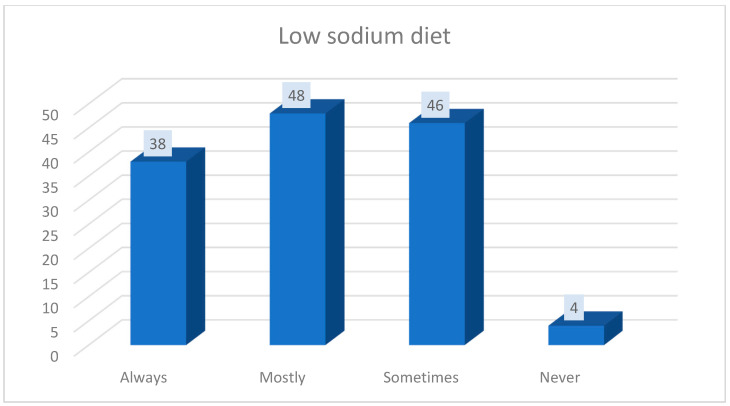
Compliance to the low-sodium diet.

**Figure 4 medicina-56-00631-f004:**
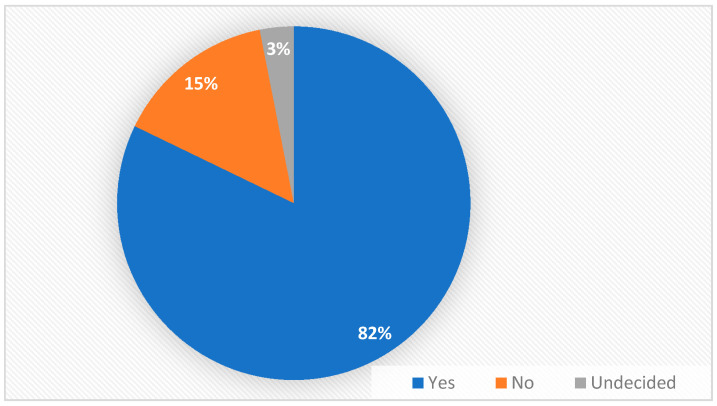
Patients’ preference for fixed dose combination.

**Figure 5 medicina-56-00631-f005:**
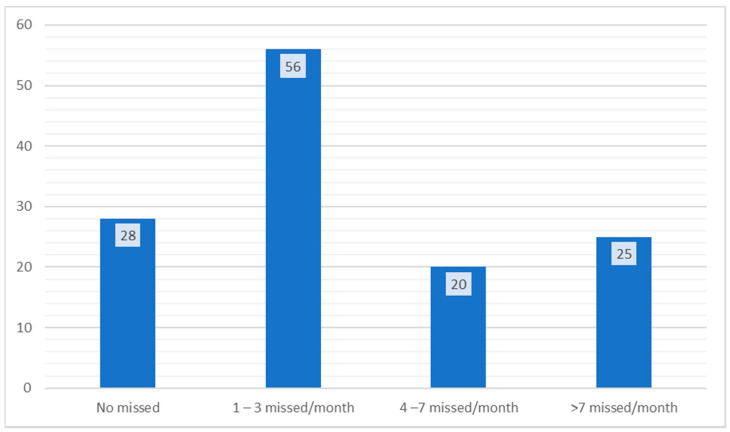
Compliance to the pharmacological treatment.

**Table 1 medicina-56-00631-t001:** Baseline characteristics of patients. BMI = body mass index.

	Group Characteristics (*n* = 129)
Gender	Female—65.89% (104)
	Male—34.10% (44)
Living environment	Urban—80.62% (104)
	Rural—19.37 (25)
Age (Mean age) = 66 ± 8 years	43–81 years old
Weight Mean BMI = 28 ± 2.4 Kg/m^2^	Normal weight—33.33% (43)
	Overweight—55.81% (72)
	Obesity grade 1—10.85% (14)
Smoking	Non-smokers—62.79% (81)
	Smokers—23.25% (30)
	Ex-smokers—13.95% (18)
Type II Diabetes	Yes—26.35% (34)
	No—73.64% (95)
Dyslipidemia	Mixed dyslipidemia—77.52% (100)
	Hypercholesterolemia—15.50% (20)
	Hypertriglyceridemia—6.97% (9)
Arterial Hypertension	Grade I—6.2% (8)
	Grade II—37.98% (49)
	Grade III—55.81% (72)

**Table 2 medicina-56-00631-t002:** Correlation between comorbidities, onset and grade of arterial hypertension and cardiologist appointment in the studied patients.

		Diabetes Mellitus	Dyslipidemia	Onset of Hypertension	Grade of Hypertension	Cardiologist Appointment	Comorbidities
Diabetes mellitus	Pearson Correlation	1	0.159	0.016	−0.083	0.194 *	0.044
	Sig. (2-tailed)		0.072	0.857	0.353	0.028	0.620
	*N*	129	129	129	129	128	129
Dyslipidemia	Pearson Correlation	0.159	1	0.137	0.172	0.309 **	0.073
	Sig. (2-tailed)	0.072		0.123	0.051	0.000	0.413
	*N*	129	129	129	129	128	129
Onset of Hypertension	Pearson Correlation	0.016	0.137	1	0.580 **	0.291 **	0.048
	Sig. (2-tailed)	0.857	0.123		0.000	0.001	0.593
	*N*	129	129	129	129	128	129
Grade of Hypertension	Pearson Correlation	–0.083	0.172	0.580 **	1	0.306 **	0.006
	Sig. (2-tailed)	0.353	0.051	0.000		0.000	0.950
	*N*	129	129	129	129	128	129
Cardiologist Appointment	Pearson Correlation	0.194 *	0.309 **	0.291 **	0.306 **	1	0.212 *
	Sig. (2-tailed)	0.028	0.000	0.001	0.000		0.016
	*N*	128	128	128	128	128	128
Comorbidities	Pearson Correlation	0.044	0.073	0.048	0.006	0.212 *	1
	Sig. (2-tailed)	0.620	0.413	0.593	0.950	0.016	
	*N*	129	129	129	129	128	129

*. Correlation is significant at the 0.05 level (2-tailed). **. Correlation is significant at the 0.01 level (2-tailed). Sig. (2-tailed) = correlation between two variables. *N* = total number.

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
