# Peer review of "Therapeutic Compliance of Patients with Arterial Hypertension in Primary Care"

_medicina, 2020, doi:10.3390/medicina56110631_

Round 1

Reviewer 1 Report

The following are suggestions for improving the paper:

Abstract

Suggested edits

Line 17……..despite all the progress made……

Line 19…. blood pressure values is the low compliance to antihypertensive treatment.

Introduction

Suggested edits

Line 45---It is recognized that arterial hypertension…..

Line 50-52-….as revealed in the Study of the Prevalence of High Blood Pressure and Cardiovascular risk assessment in Romania (SEPHAR III study ), which places……..

Line 177…..they never follow it (p = 0.001).

Line 192……..7 times a month (p = 0.01).

Line 196--………….(with a statistical significance of p=0.001).

Line 200-…..only for a limited period of time (p = 0.001).

Line 225-….significance of p=0.001)……..

Line 276-277-….and consequences of hypertension revealed the existence…….

Author Response

We are very thankful to the Editor/Reviewers for their notes; we have carefully read the comments and have revised / completed the manuscript accordingly. Our responses to both reviewers are given in a point-by-point manner below and the changes to the manuscript are highlighted in red.

We hope that in this new form the manuscript will be suitable for publication.  All the revisions that have been made are detailed, citing the line number and exact change, as requested.

Reviewer 2 Report

Title: Therapeutic Compliance of Patients with Arterial Hypertension in the Primary Care

Summary: This study's objective was to assess compliance with the treatment of patients diagnosed with arterial hypertension and monitored in a primary care office. This is a cross-sectional, retrospective study conducted one month included 129 arterial hypertension patients. Data were collected from medical files and based on the survey of 18 questions referring to hypertension, comorbidities, complication, treatment, and monitoring of the condition. The result showed that it is still a problem to follow a low sodium diet and medical regimen, and better compliance has better blood pressure control. It has verified similar findings in other studies. Insufficient diet and medication compliance are still alarmingly low.

Major:

  1. This is a small study and with self-reported compliance of diet and treatment, which is mentioned in the limitation. However, this study might be useful to guide primary care of hypertension in the specific setting of Romania.
  2. I would like to see the bar or pie graph of the compliance data, which are the key findings of this paper.
  3. I would also like to see the pie graph of the patients that responded to prefer the combo pill. This is important.

Minor:

  1. It is better to summarize the baseline descriptive data in a table with age, weight, diabetes, etc.
  2. It might be beyond the scope of this paper. However, please discuss the possible intervention specific to the Romanian population that may improve hypertension treatment compliance. These measures can be implemented in the future, and the researcher can see whether compliance improves.

Author Response

(The authors gave the same response as above.)
